# Blockwise Hadamard high-Rank Adaptation for Parameter-Efficient LLM Fine-Tuning

## Abstract

Parameter-efficient fine-tuning (PEFT) methods must be resource-efficient yet handle heterogeneous reasoning transformations, and classical low-rank adaptation (LoRA) is constrained by the nominal rank $r$. Hadamard-style extensions like HiRA raise the nominal rank but couple every update to the global energy pattern of the frozen weight matrix, while ABBA trades this inductive bias for fully learned dense intermediates. To address the limitation of global modulation, we propose Block Hadamard high-Rank Adaptation (BHRA), which partitions each weight matrix and applies HiRA-style multiplicative modulation independently within every block, preserving the PEFT parameter footprint while unlocking localized rank amplification. Our empirical analyses reveal that this blockwise design maintains rich spectra across rank budgets, mitigating the collapse induced by global modulation. Across eight commonsense reasoning tasks and two arithmetic benchmarks with Llama-3.2 1B/3B, Mistral-7B, and Gemma-2 9B, BHRA consistently surpasses strong PEFT baselines under matched parameter budgets.

## 1 Introduction

Large Language Models (LLMs) have achieved strong performance across diverse application domains, including medicine (Thirunavukarasu et al., 2023), multi-step reasoning (Wei et al., 2022), and finance (Wu et al., 2023). A common strategy for adapting LLMs to specialized domains is full fine-tuning (FFT), which updates all model parameters. However, the sheer scale of modern LLMs renders FFT computationally and storage intensive, often impractical under real-world constraints. To mitigate these costs, Parameter-Efficient Fine-Tuning (PEFT) (Ding et al., 2023) freezes the pretrained backbone and optimizes lightweight task-specific modules, thereby retaining most of the capacity of the base model while significantly reducing training overhead.

Among PEFT approaches, Low-Rank Adaptation (LoRA) (Hu et al., 2022) is particularly influential: given a base weight $W \in \mathbb{R}^{m \times n}$, LoRA introduces trainable matrices $L_1 \in \mathbb{R}^{m \times r}$ and $L_2 \in \mathbb{R}^{r \times n}$ such that the update is $\Delta W = L_1 L_2$. This construction reduces the number of trainable parameters to $(m + n)r$ while leaving inference-time computation unchanged. However, the algebraic rank of $\Delta W$ is upper bounded by $r$, so LoRA and its variants (Liu et al., 2024; Ren et al., 2024; Albert et al., 2025) may require a large $r$ to accommodate heterogeneous tasks, thereby undermining the very efficiency that motivates PEFT.

To achieve higher effective rank, Hadamard-based adapters modulate weights multiplicatively. HiRA (Huang et al., 2025) couples the frozen weight matrix $W_0$ with a low-rank factor via an element-wise product, yielding $\Delta W = W_0 \odot (BA)$, where $A$ and $B$ share LoRA's dimensionality and $\odot$ denotes the Hadamard product. In principle, this can raise the attainable rank since $\mathrm{rank}(\Delta W)$ may approach $\mathrm{rank}(W_0) \times \mathrm{rank}(BA)$. However, the modulation is global: every entry of $\Delta W$ inherits the magnitude pattern of $W_0$, limiting the ability to reallocate adaptation capacity to task-critical substructures. Alternatively, ABBA (Singhal et al., 2025) discards $W_0$ and learns two free factors $(B_1 A_1) \odot (B_2 A_2)$, improving flexibility but sacrificing the inductive bias encoded in the pretrained backbone. These approaches highlight the open question of how to retain PEFT efficiency while distributing the available rank budget more effectively than global Hadamard modulation allows.

To diagnose the underlying limitation, we examine the *stable rank* $\|\Delta W\|_F^2 / \|\Delta W\|_2^2$, a standard surrogate for effective rank. Figure 2 shows that LoRA's stable rank remains close to unity even as $r$ increases, indicating that most adaptation energy concentrates in a single dominant direction.

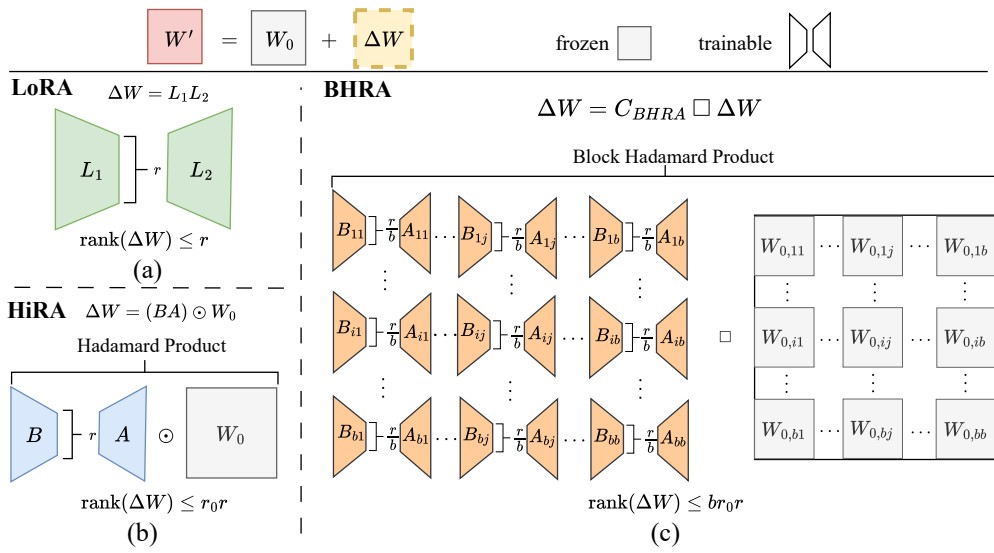

Figure 1: Illustration of BHRA compared with LoRA (Hu et al., 2022) and HiRA (Huang et al., 2025). $r_0$ and $r$ denotes the rank of $W_0$ and the total rank budget, respectively. The $b \times b$ grid indicates the block partition, and $\square$ denotes the blockwise Hadamard product.

HiRA initially achieves higher stable rank but degrades with larger budgets since the global coupling channels singular-value mass into a few directions inherited from $W_0$.

To address the limitation of global modulation, we propose Block Hadamard High-Rank Adaptation (**BHRA**), maintains substantially higher stable ranks across budgets by activating a broader set of directions while preserving parameter efficiency, as shown by the green curve in the Figure 2. As illustrated in Figure 1, BHRA partitions each weight matrix into a $b \times b$ grid and applies HiRA-style modulation independently within each block. For block $(i, j)$ with per-block rank $\frac{r}{b}$, the update is $\Delta W_{ij} = W_{0,ij} \odot (B_{ij} A_{ij})$, thereby decoupling the modulation spatially while keeping the total rank budget $r$ comparable to LoRA and HiRA. This design preserves the computational footprint of Hadamard adapters yet enables the model to deploy capacity precisely where downstream tasks demand it. As shown in Figure 3, this localized allocation translates into consistent gains across eight commonsense reasoning benchmarks on Llama-3.2 1B, with the largest improvements at higher ranks where the ability to redistribute capacity is most beneficial.

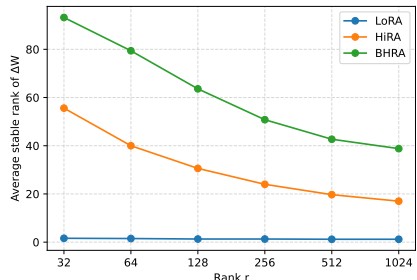

Figure 2: Average stable rank of $\Delta W$ when adapting Llama-3.2 1B to commonsense reasoning. BHRA maintains a substantially larger effective rank across identical rank budgets $r$.

Our contributions are threefold:

- We provide quantitative analyses of Hadamard-style adapters, demonstrating global modulation collapses the effective rank of $\Delta W$ while blockwise modulation maintains diverse directions.

- To address the limitation of global modulation, we propose Block Hadamard High-Rank Adaptation (**BHRA**), a block-partitioned HiRA variant that preserves PEFT efficiency yet expands the attainable rank under a fixed parameter budget.

- We conduct extensive experiments on multiple benchmarks and demonstrates the effectiveness of BHRA against representative PEFT baselines.

## 2 RELATED WORK

**Low-rank adaptation**. LoRA Hu et al. (2022) parameterizes the update as a low-rank decomposition $\Delta W = BA$, freezing $W_0$ and training only $A, B$. It achieves large parameter and memory savings with negligible inference overhead and has the common budget or implementation baseline for PEFT. Building on this, AdaLoRA Zhang et al. (2023) adaptively allocates the rank budget across layers by importance, improving utilization under the same total budget. DoRA Liu et al. (2024) decomposes each pretrained weight into magnitude and direction and applies LoRA only to the directional component, narrowing the gap to full fine-tuning without extra inference cost. GraLoRA Jung et al. (2025) introduces granularity: it partitions a weight matrix into sub-

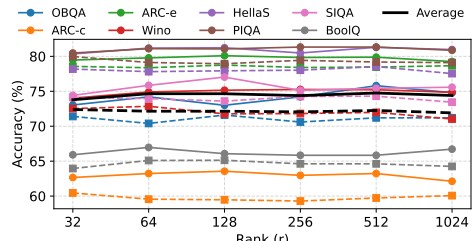

Figure 3: Performance of BHRA (solid) and HiRA (dashed) of Llama-3.2 1B on eight commonsense reasoning datasets using different HiRA configurations.

blocks and matches a tiny LoRA to each block, mitigating structural bottlenecks and boosting expressivity at essentially the same parameter and FLOPs scale as standard LoRA.

A parallel line raises effective rank via Hadamard product, using the inequality $\mathrm{rank}(O_1 \odot O_2) \leq \mathrm{rank}(O_1) \times \mathrm{rank}(O_2)$ Million (2007). HiRA Huang et al. (2025) writes $\Delta W = W_0 \odot (BA)$, leveraging the typically high rank of $W_0$ to exceed LoRA's limit while keeping LoRA-level parameter cost. ABBA Singhal et al. (2025) fully decouples from $W_0$ by learning two low-rank factors and taking their Hadamard product, $\Delta W = (B_1 A_1) \odot (B_2 A_2)$, yielding higher expressivity under the same budget. In this paper, we introduce BHRA, the first blockwise Hadamard formulation of $\Delta W$ for PEFT via block Hadamard product: partition $W_0$ and apply HiRA-style modulation independently per block, $\Delta W_{ij} = \Delta W_{0,ij} \odot (B_{ij} A_{ij})$. BHRA (1) strictly generalizes HiRA, which is the $1 \times 1$ case, (2) raises attainable effective rank by per-block bounds $\mathrm{rank}(\Delta W_{ij}) \leq \mathrm{rank}(W_{0,ij}) \times \mathrm{rank}(B_{ij} A_{ij})$, and (3) preserves HiRA-style parameters and FLOPs while adding spatial controllability absent in global Hadamard or purely additive block schemes.

**Other PEFT**. Beyond low-rank updates, adapter tuning Houlsby et al. (2019) freezes the backbone and inserts small bottlenecks. The original Adapter layers establish the template for parameter sharing across tasks. Prompt tuning Lester et al. (2021) and prefix tuning Li & Liang (2021) keeps all weights fixed and instead learns continuous prompts or layer-wise key or value prefixes, with Prefix-Tuning targeting generation and Prompt Tuning becoming competitive with full fine-tuning as model size scales to billions. These categories primarily optimize storage, compute, and task compositionality, and are complementary to BHRA, which instead targets higher effective rank of $\Delta W$ via blockwise Hadamard modulation under LoRA-level parameter and FLOPs.

## 3 METHODOLOGY

In this section, we first revisit the hadamard-style adaptation, and then introduce Block Hadamard high-Rank Adaptation (BHRA). Next, we provide a theoretical analysis of the expressive power of BHRA. Finally, we present the training and inference efficiency and gradient analysis for BHRA.

**Revisiting Hadamard-Style Adaptation**. Let $W_0 \in \mathbb{R}^{m \times n}$ denote the frozen weight matrix of a linear layer and let $r$ be the available rank budget. Following Hu et al. (2022), LoRA introduces trainable factors $L_1 \in \mathbb{R}^{m \times r}$ and $L_2 \in \mathbb{R}^{r \times n}$ such that

$$\Delta W_{\mathrm{LoRA}} = L_1 L_2, \tag{1}$$

and therefore $\mathrm{rank}(\Delta W_{\mathrm{LoRA}}) \leq r$. HiRA (Huang et al., 2025) retains this low-rank scaffold but modulates the update multiplicatively with the pretrained weights:

$$\Delta W_{\mathrm{HiRA}} = W_0 \odot (BA), \tag{2}$$

where $\odot$ denotes the Hadamard (element-wise) product. The attainable rank of $\Delta W_{\mathrm{HiRA}}$ follows from the classical Hadamard product inequality as follows.

**Lemma 3.1** (Hadamard rank bound (Million, 2007)). *For any matrices $O_1$ and $O_2$ of the same size,*

$$\mathrm{rank}(O_1 \odot O_2) \leq \mathrm{rank}(O_1)\,\mathrm{rank}(O_2). \tag{3}$$

Table 1: Comparison of Hadamard-based adapters under an equal parameter/rank budget.

| Method | Trainable parameters | Rank upper bound | Dependence on $W_0$ |
|--------|---------------------|------------------|---------------------|
| LoRA | $r(m+n)$ | $r$ | None |
| HiRA | $r(m+n)$ | $r_0 r$ | Global modulation by $W_0$ |
| ABBA | $r(m+n)$ | $r^2$ | None; dense intermediates |
| BHRA | $r(m+n)$ | $br_0 r$ | Blockwise modulation by $W_0$ |

Applying Lemma 3.1 with $O_1 = W_0$ and $O_2 = BA$ yields

$$\mathrm{rank}(\Delta W_{\mathrm{HiRA}}) \leq \mathrm{rank}(W_0)\,\mathrm{rank}(BA) \leq \mathrm{rank}(W_0)\,r, \tag{4}$$

showing that HiRA multiplies LoRA's rank bound by $\mathrm{rank}(W_0)$ while preserving the trainable parameter count $r(m+n)$.

Recently, to address the reliance on $W_0$ in HiRA, ABBA (Singhal et al., 2025) learns two independent low-rank products whose Hadamard combination forms the update, thereby discarding the dependence on $W_0$ but requiring dense $m \times n$ intermediates. Alternatively, BHRA aims to combine HiRA's inductive bias with ABBA's flexibility while keeping a LoRA-level parameter footprint.

In Table 1, we summarize the principal Hadamard-style adapters under a shared rank budget $r$ for an $m \times n$ layer. LoRA trains $r(m+n)$ parameters and its update rank is bounded by $r$. HiRA multiplies this bound by $\mathrm{rank}(W_0) = r_0$. ABBA reaches $r^2$ by learning two dense intermediates. BHRA redistributes the rank budget across blocks of $W_0$, retaining the same parameter count while enabling localized amplification.

**Block-Hadamard High-Rank Adaptation (BHRA)**. BHRA partitions $W_0$ into a $b \times b$ grid of disjoint blocks. Let each block share dimensions $m/b$ by $n/b$ for clarity (other partitions follow analogously), and denote block $(i, j)$ by $W_{0,ij} \in \mathbb{R}^{(m/b) \times (n/b)}$. Within every block we allocate low-rank factors

$$B_{ij} \in \mathbb{R}^{(m/b) \times (r/b)}, \qquad A_{ij} \in \mathbb{R}^{(r/b) \times (n/b)}, \tag{5}$$

whose product $C_{ij} = B_{ij} A_{ij}$ has rank at most $r/b$. Arranging these factors into a block matrix produces a "capacity" tensor

$$C_{\mathrm{BHRA}} = \begin{bmatrix} C_{11} & \cdots & C_{1b} \\ \vdots & \ddots & \vdots \\ C_{b1} & \cdots & C_{bb} \end{bmatrix} \in \mathbb{R}^{m \times n}. \tag{6}$$

The BHRA update is obtained by a blockwise Hadamard product between this learned capacity and the frozen weights:

$$\Delta W_{\mathrm{BHRA}} = C_{\mathrm{BHRA}} \square W_0, \tag{7}$$

where $\square$ applies the element-wise Hadamard product inside each block (Günther & Klotz, 2012). Setting $b = 1$ recovers HiRA. The trainable parameter count remains $r(m+n)$ because the $b^2$ blocks each store a rank-$\frac{r}{b}$ pair.

This construction mirrors the schematic in Figure 1: orange low-rank LoRA pairs $(B_{ij}, A_{ij})$ tile the matrix, the grey tiles represent the corresponding submatrices of $W_0$, and the block Hadamard operator combines them to form $\Delta W_{\mathrm{BHRA}}$. Localizing the modulation allows BHRA to amplify rank in every block without materializing dense intermediates.

**Expressive Power of BHRA**. We begin by bounding the rank of the learned capacity that mediates BHRA's update. Let $r_0 = \mathrm{rank}(W_0)$.

**Lemma 3.2** (Rank of the BHRA capacity). *For a $b \times b$ partition with uniform rank budget, $\mathrm{rank}(C_{BHRA}) \leq br$.*

*Proof.* Fix a row $i$. The horizontal concatenation $[C_{i1} \cdots C_{ib}]$ consists of $b$ blocks, each with rank at most $r/b$. The rank of a horizontal concatenation is no larger than the sum of the ranks of its constituents, so the $i$-th row block has rank at most $r$. Stacking the $b$ row blocks vertically increases the rank by at most $r$ per block, yielding $\mathrm{rank}(C_{\mathrm{BHRA}}) \leq br$. $\square$

With this capacity bound in hand, combining Lemma 3.2 with Lemma 3.1 yields the desired rank guarantee for BHRA as follows.

**Proposition 3.1** (BHRA rank upper bound). *Let* $r_0 = \text{rank}(W_0)$. *Then*

$$\text{rank}(\Delta W_{BHRA}) = \text{rank}(C_{BHRA} \square W_0) \leq \text{rank}(C_{BHRA}) \text{rank}(W_0) \leq b r_0 r. \tag{8}$$

The factor $b$ reflects the number of block rows; non-square grids simply replace it with the number of row partitions. Thus BHRA scales the HiRA bound by the number of block slices while preserving the parameter footprint $r(m + n)$. Empirically, pretrained blocks often exhibit high local rank, so the $b$-fold amplification translates into near-full-rank updates under the same budget. We analyze parallel bounds for other Hadamard-style adapters next and include full derivations in Appendix A.

*Comparison with HiRA*. For HiRA, letting $\text{rank}(W_0) = r_0$ and writing $\Delta W_{\text{HiRA}} = W_0 \odot (BA)$ with $B \in \mathbb{R}^{m \times r}$ and $A \in \mathbb{R}^{r \times n}$, the Hadamard rank inequality ensures

$$\text{rank}(\Delta W_{\text{HiRA}}) = \text{rank}(W_0 \odot (BA)) \leq r_0 \text{rank}(BA) \leq r_0 r.$$

Coupled with Proposition 3.1, which yields $\text{rank}(\Delta W_{\text{BHRA}}) \leq b r_0 r$, we see that BHRA scales HiRA's attainable rank by a factor of $b$ while preserving the identical parameter budget $r(m + n)$. When $b = 1$, the two bounds coincide, recovering HiRA.

**Training and Inference Efficiency**. First, we analyze the expected computational cost of LoRA in terms of FLOPs. LoRA adapts each linear layer with a pair of rank-$r$ GEMMs: the projection $Z = AX$ costs $(2n - 1)rT$ FLOPs and the reconstruction $Y = BZ$ costs $(2r - 1)mT$, yielding

$$\text{FLOPs}_{\text{LoRA}} = (2n - 1)rT + (2r - 1)mT = 2r(m + n)T - (r + m)T, \tag{9}$$

while caching only the $rT$ activations in $Z$. HiRA keeps these two multiplications but gates the update with a mask $W_0 \odot (BA)$ that we materialize once, so its adapter overhead stays at $\approx 2r(m + n)T$ with the same activation footprint.

BHRA partitions the layer into $b \times b$ blocks. Summing the per-block projections, reconstructions, mask multiplications, and refreshes gives

$$\text{FLOPs}_{\text{BHRA}}^{(\text{train})} = 2r(m + n)T - 2brT + \frac{mn}{b^2}T + \frac{2mnr}{b}, \tag{10}$$

which collapses to the HiRA expression when $b = 1$. Before deployment we fold the learned masks $H_{ij}$ into $W_0$, leaving inference with the same two rank-$r$ GEMMs $(2n - 1)rT + (2m - 1)rT$ and a static buffer of size $mn/b^2$ in addition to the usual $rT$ cache. Please find the detailed analysis of the trade-off in Appendix B.

**Gradient Analysis**. Following the exposition of Huang et al. (2025), let a linear layer produce $z = (W_0 + \Delta W)x$ and incur loss $\mathcal{L}$ with residual $g = \nabla_z \mathcal{L}$. The gradient with respect to the update parameters factors through $G = gx^\top$.

In LoRA the two low-rank factors are differentiated as $\nabla_A \mathcal{L} = B^\top G$ and $\nabla_B \mathcal{L} = GA^\top$, so the gradients are completely agnostic to the pretrained weights $W_0$. HiRA inserts $W_0$ multiplicatively and the gradients become $\nabla_A \mathcal{L} = B^\top (W_0 \odot G)$ and $\nabla_B \mathcal{L} = (W_0 \odot G) A^\top$, revealing that HiRA leverages the structure already encoded in $W_0$ to steer the update directions.

BHRA preserves this inductive bias while localising it. Partition the residual and input as $g = [g_1^\top, \ldots, g_b^\top]^\top$ and $x = [x_1^\top, \ldots, x_b^\top]^\top$, so the blockwise gradients factor through $G_{ij} = g_i x_j^\top$. For block $(i, j)$,

$$\nabla_{A_{ij}} \mathcal{L} = B_{ij}^\top (W_{0,ij} \odot G_{ij}), \qquad \nabla_{B_{ij}} \mathcal{L} = (W_{0,ij} \odot G_{ij}) A_{ij}^\top.$$

Thus each block is guided only by its corresponding slice $W_{0,ij}$, preventing globally small entries of $W_0$ from suppressing gradients elsewhere while retaining the beneficial alignment that differentiates HiRA from LoRA.

## 4 EXPERIMENTS

In this section, we evaluate BHRA on commonsense and arithmetic reasoning tasks. We will introduce the datasets, experimental settings and results on these tasks.

## 4.1 DATASETS

**Commonsense reasoning**. We utilize eight sub-tasks with predefined training and testing datasets (Hu et al., 2023), combining 170,420 query-answer pairs for fine-tuning LLMs and selecting 120 random entries as a validation set. The eight sub-tasks includes BoolQ (Clark et al., 2019), PIQA (Bisk et al., 2020), SIQA (Sap et al., 2019), HellaSwag (Zellers et al., 2019), WinoGrande (Sakaguchi et al., 2021), ARC-Challenge and ARC-Easy (Clark et al., 2018), and OBQA Mihaylov et al. (2018). We evaluate performance on each dataset independently to capture task-specific generalization.

**Arithmetic reasoning**. We fine-tune Mistral-7B (Jiang et al., 2023) and Gemma-2 9B (Team et al., 2024) on a 50K-sample subset of MetaMathQA (Yu et al., 2023), and evaluate on MATH (Hendrycks et al., 2021) and GSM8K (Cobbe et al., 2021), reporting exact-match accuracy, consistent with prior work (Singhal et al., 2025).

For both tasks, We insert LoRA adapters into all attention projections including query, key, value, and output as well as feedforward network layers.

## 4.2 EXPERIMENTAL SETTINGS

**Baselines**. We compare against representative PEFT methods under matched parameter budgets: LoRA (Hu et al., 2022), DoRA (Liu et al., 2024), HiRA (Huang et al., 2025), ABBA (Singhal et al., 2025). Our experiments use the Llama-3.2-1B and Llama-3.2-3B (Grattafiori et al., 2024), Mistral-7B (Jiang et al., 2023) and Gemma-2 9B (Team et al., 2024) open-source LLMs.

**Metrics**. For commonsense reasoning we adopt accuracy as the primary metric, consistent with Huang et al. (2025); Singhal et al. (2025). Given a model completion, we apply a task-specific post-processing step: the generated text is scanned for canonical answer tokens (e.g., true"/false" for BoolQ, option letters for PIQA, SIQA, ARC, and OBQA). The first matched token is treated as the prediction; if no valid token is detected the response is marked incorrect. Accuracy is computed separately for each dataset and we report the macro-average over all eight tasks to control for dataset size imbalance. Arithmetic benchmarks follow the exact-match protocol of GSM8K/MATH. Outputs are normalized by stripping punctuation, lowercasing, and resolving verbal numbers to their numeric forms so that mathematically equivalent answers are aligned. A prediction counts as correct only when the normalized string matches the reference exactly.

**Implementation details**. Following the identical training setup to Huang et al. (2025); Singhal et al. (2025) except learning rate adjustments, we implement BHRA on all reasoning tasks with total rank settings $r_{tot} = 32$. We train for 2 epochs and 1 epoch for the commonsense reasoning and arithmetic reasoning tasks, respectively. Results are reported as the mean over 5 random seeds. These choices mirror the HiRA setup for fair comparison. In BHRA, we set $p = q = b$ and $r_b = \frac{r_{tot}}{b}$ to exactly matches LoRA/HiRA parameter counts and FLOPs. We use this setting in all main comparisons. We adopt LoRA-style scaling $\alpha = r_{tot}$ and standard initialization that yields zero initial update (Hu et al., 2022), matching HiRA's practice to preserve the base model at step 0. We use AdamW with learning rate 0.002 and 100 warm-up steps as in (Singhal et al., 2025). As with HiRA and ABBA, we support pre-compute and add $\Delta W$ to $W_0$, yielding zero inference overhead beyond the base model. Please find the detailed implementation in Appendix C.

## 4.3 RESULTS ON COMMONSENSE REASONING

As shown in Table 2, on Llama-3.2-1B, BHRA attains an average accuracy of 73.76, improving over LoRA (70.75%) by +3.01% points and over HiRA (72.40%) by +1.36% points, while remaining within 0.03% of ABBA (73.79%). On Llama-3.2-3B, BHRA reaches 84.52%, which is +2.73% above LoRA and +0.69% above HiRA, and is 0.24% below ABBA. Gains are strongest on tasks such as ARC-c, ARC-e, Wino, and BoolQ, where models benefit from combining diverse local transformations. The blockwise Hadamard update increases local effective rank without increasing the parameter or FLOP budget, and it allows different regions of the weight matrix to specialize. This combination improves coverage of subspaces that matter for multi-facet commonsense reasoning.

Table 2: Comparison of multiple fine-tuning methods on Llama-3.2 1B and 3B across eight common-sense reasoning datasets. Best results among PEFT methods are in **bold**.

| Model | Method | # Params | Accuracy (↑) | | | | | | | | |
|---|---|---|---|---|---|---|---|---|---|---|---|
| | | | OBQA | ARC-c | ARC-e | Wino | HellaS | PIQA | SIQA | BoolQ | Avg. |
| Llama-3.2 1B | FFT | 1.24 B | 74.00 | 62.05 | 78.63 | 74.79 | 79.63 | 80.62 | 75.37 | 63.77 | 73.61 |
| | LoRA | 22.54 M | 68.48 | 58.91 | 76.67 | 71.95 | 75.45 | 77.79 | 72.94 | 63.82 | 70.75 |
| | DoRA | 22.92 M | 70.00 | 59.57 | 77.50 | 72.70 | 75.46 | 78.42 | 73.14 | 63.71 | 71.31 |
| | HiRA | 22.54 M | 71.40 | 60.49 | 78.56 | 72.52 | 78.19 | 79.97 | 74.12 | 63.95 | 72.40 |
| | ABBA | 22.54 M | **73.60** | 61.52 | 79.04 | **74.19** | **81.87** | **81.12** | 74.00 | 65.02 | 73.79 |
| | BHRA | 22.54 M | 73.04 | **62.66** | **79.47** | 74.02 | 80.35 | 80.47 | **74.41** | **65.91** | **73.79** |
| Llama-3.2 3B | FFT | 3.21 B | 85.00 | 78.81 | 90.00 | 86.55 | 93.14 | 87.25 | 81.49 | 73.58 | 84.48 |
| | LoRA | 48.63 M | 82.27 | 76.19 | 87.52 | 84.00 | 91.25 | 84.38 | 79.06 | 69.66 | 81.79 |
| | DoRA | 49.40 M | 82.80 | 76.59 | 89.05 | 85.86 | 92.71 | 85.92 | 80.82 | 71.02 | 83.10 |
| | HiRA | 48.63 M | 83.52 | 77.13 | 89.38 | 85.30 | 92.91 | 86.49 | 80.71 | 72.68 | 83.51 |
| | ABBA | 48.63 M | 84.76 | 78.24 | 89.76 | 86.28 | **93.51** | 86.91 | 80.82 | 73.63 | 84.24 |
| | BHRA | 48.63 M | **85.16** | **78.48** | **90.03** | **86.49** | 93.37 | **87.03** | **81.28** | **74.35** | **84.52** |

Table 3: Comparison of multiple fine-tuning methods on Mistral-7B and Gemma-2 9B across arithmetic reasoning benchmarks. Best results among PEFT methods are in **bold**.

| Method | Mistral-7B | | | Gemma-2 9B | | |
|---|---|---|---|---|---|---|
| | # Params | GSM8K (↑) | MATH (↑) | # Params | GSM8K (↑) | MATH (↑) |
| FFT | 7.24 B | 67.74 | 19.62 | 9.24 B | 79.28 | 39.89 |
| LoRA | 83.88 M | 61.94 | 15.98 | 108.04 M | 76.19 | 36.56 |
| DoRA | 85.26 M | 65.73 | 19.02 | 109.88 M | 76.91 | 38.05 |
| HiRA | 83.88 M | 66.29 | 17.77 | 108.04 M | 78.74 | 38.11 |
| ABBA | 83.88 M | 66.57 | 18.03 | 108.04 M | 78.70 | **38.80** |
| BHRA | 83.88 M | **66.64** | **20.07** | 108.04 M | **78.98** | 38.13 |

## 4.4 RESULTS ON ARITHMETIC REASONING

As shown in Table 3, on Mistral-7B, BHRA reaches 66.64% on GSM8K and 20.07% on MATH. This improves over LoRA by 4.70% and 4.09%, over HiRA by 0.35% and 2.30%, and over ABBA by 0.07% and 2.04%. The larger gains on MATH indicate that blockwise Hadamard modulation raises useful local rank for long, stepwise derivations. These two benchmarks are standard multi-step arithmetic tests in the literature. On Gemma-2 9B, BHRA attains 78.98% on GSM8K and 38.13% on MATH. Relative to LoRA, the improvements are 2.79% and 1.57%. BHRA edges HiRA on GSM8K by 0.24% and is effectively tied on MATH with a 0.02% point lead, while it is 0.28% above ABBA on GSM8K and 0.67% below on MATH.

## 5 ANALYSIS

### 5.1 SINGULAR VALUE STRUCTURE OF FULL FINE-TUNING, HiRA, AND BHRA

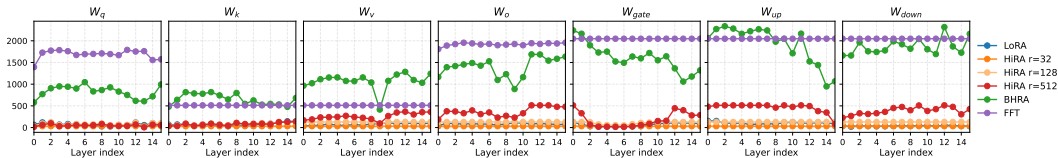

Figure 4: Count of singular values exceeding 1% of the layer-wise maximum for FFT, LoRA, HiRA, and BHRA.

Figure 4 reports the number of singular values above the 1% energy threshold for each transformer projection. Full fine-tuning (FFT) activates the richest spectrum throughout the stack, and BHRA closely tracks this envelope, particularly in the FFN up/down projections where its counts remain high even in deeper layers. HiRA exhibits a clear dependence on the nominal rank: at $r = 32$ it delivers

only a modest lift over LoRA, $r = 128$ roughly doubles the active directions in the feed-forward pathway, and $r = 512$ pushes the spectrum further but still falls short of the blockwise coverage achieved by BHRA. LoRA remains almost flat across layers, reflecting the severe bottleneck imposed by its single low-rank factorization.

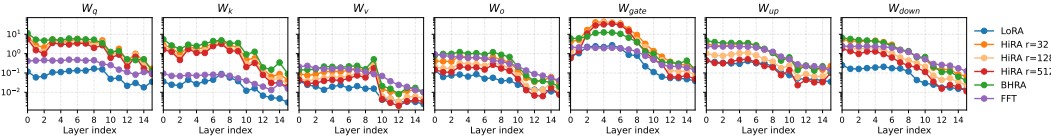

Figure 5: Layer-wise sum of squared singular values for FFT, LoRA, HiRA, and BHRA.

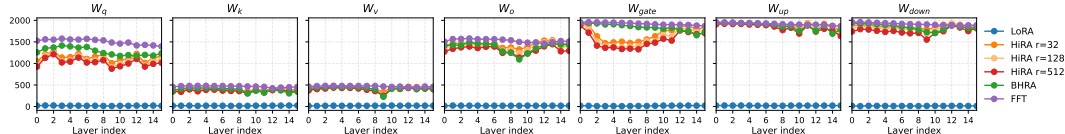

Figure 6: Effective rank across layers for FFT, LoRA, HiRA, and BHRA.

The aggregate singular-value energy in Figure 5 mirrors this stratification. BHRA preserves nearly all of the FFT energy in every projection, whereas HiRA recovers progressively more mass as the rank increases from 32 to 128 and 512, yet each variant still trails BHRA in the deeper FFN blocks. LoRA consistently captures the least energy, underscoring that a global low-rank adapter fails to populate directions that dominate the spectrum. This indicates that LoRA stays rank-deficient in every module.

Effective-rank trends in Figure 6 reinforce the conclusion. BHRA maintains an entropy-based rank profile that is indistinguishable from full fine-tuning across the network. HiRA's curves again separate by nominal rank: the $r = 32$ model plateaus early, the $r = 128$ setting narrows the gap through the middle of the stack, and $r = 512$ approaches BHRA only in the lower layers while still lagging in the upper decoder blocks. The stability analysis in Figure 2 confirms that this hierarchy persists when sweeping the target rank: BHRA keeps a higher stable rank than HiRA across the sweep and LoRA remains nearly flat near one.

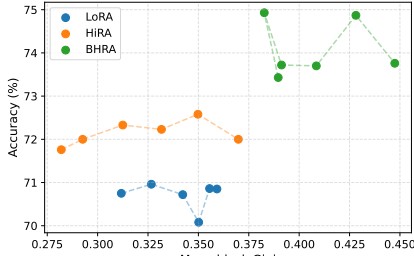

Figure 7: Accuracy on the eight commonsense tasks versus mean block Gini of the learned adapters (Llama-3.2-1B, $r_b \times b = 32$). Greater block heterogeneity correlates with higher accuracy, and BHRA dominates both LoRA and HiRA in this trade-off.

Together, these spectral diagnostics explain the accuracy hierarchy observed in Table 2 and Table 3: BHRA's blockwise capacity unlocks rich, layer-local updates that emulate the representational flexibility of full fine-tuning, whereas HiRA and LoRA remain constrained by their limited ability to populate the singular spectrum.

## 5.2 ANALYSIS OF BLOCK HETEROGENEITY

Figure 7 plots macro accuracy against the mean block Gini of the learned updates. The three PEFT baselines form clearly separated clusters: LoRA achieves the lowest heterogeneity and correspondingly lags in accuracy, HiRA raises both quantities modestly, while BHRA consistently occupies the upper-right region. This monotonic trend supports our claim that BHRA's blockwise Hadamard modulation induces richer intra-layer variation captured by higher block Gini coefficient, which in turn translates into stronger commonsense performance under a fixed parameter budget.

## 5.3 CHOICE OF $r_b$ FOR BHRA

Under a fixed budget $r_b \times b = 32$, we sweep $r_b \in \{1, 2, 4, 8, 16, 32\}$ with $b = 32/r_b$ on commonsense reasoning tasks shown in Figure 8 and arithmetic tasks (Please find details in Appendix D). Across both settings, $r_b = 4, b = 8$ is consistently on the accuracy maximum while remaining stable across

the benchmark models and datasets; very small $r_b$ with many tiny blocks underfits, where larger $r_b$ with few blocks loses block diversity and slightly degrades performance. Therefore, we set $r_b = 4, b = 8$ as a balanced point between per-block expressivity and block diversity under the same parameter budget.

## 5.4  ANALYSIS OF PLACEMENT OF BHRA IN TRANSFORMERS

Table 4: Impact of selectively fine-tuning individual transformer components - Key, Query, Value, Output, Up, Gate, and Down projections, with BHRA on Llama-3.2-1B.

| Component | OBQA | ARC-c | ARC-e | Wino | HellaS | PIQA | SIQA | BoolQ | Avg. |
|-----------|-------|-------|-------|-------|--------|-------|-------|-------|-------|
| All | 73.04 | 62.66 | 79.47 | 74.02 | 80.35 | 80.47 | 74.41 | 65.91 | 73.79 |
| FFN | 73.10 | 62.37 | 79.48 | 73.90 | 79.60 | 81.04 | 74.16 | 65.79 | 73.68 |
| Down | 68.80 | 59.00 | 76.70 | 69.97 | 73.44 | 77.48 | 72.68 | 64.50 | 70.32 |
| Gate | 65.73 | 57.17 | 76.25 | 68.48 | 71.32 | 77.37 | 71.60 | 64.18 | 69.01 |
| Up | 67.80 | 63.29 | 67.57 | 73.37 | 71.85 | 75.66 | 75.17 | 68.21 | 67.12 |
| O | 66.52 | 57.13 | 75.29 | 68.92 | 74.27 | 78.16 | 72.36 | 63.60 | 69.46 |
| QKV | 66.68 | 57.49 | 75.64 | 70.73 | 75.62 | 78.01 | 73.19 | 62.31 | 69.96 |
| V | 57.95 | 50.80 | 69.98 | 64.39 | 67.85 | 76.43 | 67.90 | 60.96 | 64.70 |
| K | 58.50 | 48.98 | 69.70 | 63.63 | 63.21 | 74.08 | 65.90 | 57.32 | 62.66 |
| Q | 57.20 | 48.55 | 67.06 | 65.07 | 58.54 | 73.48 | 64.69 | 60.99 | 61.95 |

Table 4 quantifies how BHRA behaves when restricted to individual projection matrices inside the Llama-3.2-1B transformer. Adapting every linear submodule ("All") achieves the strongest average accuracy (73.79) and principally serves as an upper bound. Limiting BHRA to the feed-forward pathway (FFN: Up, Gate, and Down) nearly matches this ceiling at 73.68 average, underscoring that most of the attainable improvements arise from the MLP stack. Among single-component interventions, the Down projection delivers the largest gain (70.32 average), followed by the Gate (69.01) and Up (67.12) branches. Each of these submodules modulates feature transformation and routing, allowing BHRA to inject diverse block-specific updates. In contrast, up-

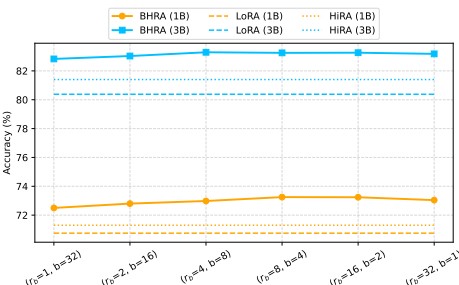

Figure 8: The performance of BHRA, LoRA and HiRA under fixed budget ($r_b \times b = 32$) on commonsense reasoning tasks.

dating the attention projections alone is markedly less effective: the Output projection reaches 69.46, and adapting only Q, K, or V hovers near 62–65. Even treating QKV jointly recovers to just 69.96, still below the FFN-focused variants. These trends align with the functional roles of the submodules: attention weights primarily steer token-to-token interactions, whereas the feed-forward projections reshape hidden states and thus provide a richer canvas for blockwise Hadamard modulation.

## 6  CONCLUSION

In this paper, we propose Block Hadamard high-Rank Adaptation (BHRA), which mitigates the rank bottlenecks of classical low-rank and Hadamard-style adapters by partitioning pretrained weights and applying HiRA-style modulation locally. This blockwise formulation preserves LoRA-level efficiency while expanding the attainable rank of $\Delta W$, as corroborated by stable-rank and spectral analyses. Across eight commonsense reasoning tasks and two arithmetic benchmarks on Llama-3.2 1B/3B, Mistral-7B, and Gemma-2 9B, BHRA consistently outperforms representative PEFT baselines under matched parameter and FLOP budgets. Ablation studies show that moderate block counts with modest per-block rank offer a resilient accuracy–efficiency trade-off. These results highlight block-level heterogeneity as a key determinant of high-utility PEFT and establish BHRA as a practical, theoretically grounded alternative to existing Hadamard adapters. Future work will pursue data-driven block partitions, multi-modal extensions, and integration with adaptive scheduling or continual learning.

## 7 REPRODUCIBILITY STATEMENT

We have taken several steps to ensure reproducibility of our work. Details of the model architecture, BHRA configurations and training hyperparameters are provided in Section 4 and Appendix C. The datasets used for training and evaluation are publicly available and fully described in Section 4.1. We also provide the directory of the source code and scripts for reproducing all experiments in the supplementary materials. This includes implementations of our BHRA, training scripts, and configuration files.

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

## A  DETAILED ANALYSIS OF RANK UPPER BOUNDS

**HiRA rank upper bound.**   Let $W_0 \in \mathbb{R}^{m \times n}$ with $\mathrm{rank}(W_0) = r_0$. HiRA parameterises the update as $\Delta W_{\text{HiRA}} = W_0 \odot (BA)$ with $B \in \mathbb{R}^{m \times r}$ and $A \in \mathbb{R}^{r \times n}$. Using the Hadamard rank inequality,

$$\mathrm{rank}(\Delta W_{\text{HiRA}}) = \mathrm{rank}\big(W_0 \odot (BA)\big) \leq r_0 \,\mathrm{rank}(BA) \leq r_0 r,$$

so the HiRA update obeys

$$\boxed{\mathrm{rank}(\Delta W_{\text{HiRA}}) \leq r_0 r}.$$

**ABBA rank upper bound.**   ABBA learns two independent low-rank products, $P = B_1 A_1$ and $Q = B_2 A_2$, each of rank at most $r$. Their Hadamard combination expands as

$$\Delta W_{\text{ABBA}} = P \odot Q = \sum_{i=1}^{r} \sum_{j=1}^{r} (b_{1,i} \odot b_{2,j}) \,(a_{1,i} \odot a_{2,j})^\top, \tag{11}$$

where $b_{1,i}$ denotes the $i$-th column of $B_1$, etc. Consequently $\mathrm{rank}(\Delta W_{\text{ABBA}}) \leq r^2$ irrespective of $W_0$; the attainable rank is capped by $r_{\text{tot}}^2/4$ even when the layer dimension is much larger.

**BHRA rank upper bound.**   For BHRA we reuse the block construction described earlier; The learned capacity $C_{\text{BHRA}}$ satisfies Lemma 3.2, namely $\mathrm{rank}(C_{\text{BHRA}}) \leq br$ for a uniform $b \times b$ partition. Combining Lemma 3.2 with Lemma 3.1 yields Proposition 3.1:

$$\mathrm{rank}(\Delta W_{\text{BHRA}}) = \mathrm{rank}(C_{\text{BHRA}} \square W_0) \leq b \, r_0 r.$$

As before the ambient dimension enforces $\mathrm{rank}(\Delta W_{\text{BHRA}}) \leq \min\{m, n\}$, so altogether

$$\boxed{\mathrm{rank}(\Delta W_{\text{BHRA}}) \leq b r_0 r}.$$

The factor $b$ reflects the number of block rows; non-square partitions simply replace it with the number of row groups.

**Relationship.**   Comparing the bounds

$$\mathrm{rank}(\Delta W_{\text{HiRA}}) \leq r_0 r, \qquad \mathrm{rank}(\Delta W_{\text{BHRA}}) \leq b r_0 r,$$

shows that BHRA scales HiRA's rank budget by a factor of $b$ while keeping the same parameter count $r(m + n)$. When $b = 1$ the two bounds coincide, recovering HiRA.

## B  DETAILED ANALYSIS OF TRADE-OFF

We first quantify the training cost of LoRA and HiRA before detailing BHRA and its relation to GraLoRA. All derivations assume a mini-batch with sequence length $T$.

**LoRA overhead recap.**   LoRA performs the projection $Z = AX$ with cost $(2n - 1)rT$ FLOPs followed by the reconstruction $Y = BZ$ with cost $(2r - 1)mT$. Summing the two gives

$$\mathrm{FLOPs}_{\text{LoRA}} = (2n - 1)rT + (2r - 1)mT = 2r(m + n)T - (r + m)T. \tag{12}$$

No Hadamard modulation is involved, and the only activations cached for backpropagation are the rank-$r$ projections $Z$, totalling $rT$ elements.

**HiRA overhead recap.**   HiRA retains LoRA's two GEMMs but multiplies the update by $W_0$ element-wise. The projection and reconstruction costs remain $(2n - 1)rT$ and $(2m - 1)rT$. The Hadamard modulation introduces an $mnT$ element-wise product with the same shape as the base model matmul $W_0 X$. We include this term in the total FLOP expression above, but emphasize that it matches the dense baseline cost already incurred by the frozen layer. Hence, when comparing adapter overheads we focus on the extra low-rank multiplications, which remain $O(r(m + n)T)$ as in LoRA. Collecting the adapter-specific terms yields

$$\mathrm{FLOPs}_{\text{HiRA}}^{(\text{adapter})} = (2n - 1)rT + (2m - 1)rT \approx 2r(m + n)T. \tag{13}$$

HiRA therefore matches LoRA's asymptotic adapter cost while keeping the same $rT$ activation cache.

**BHRA overhead.**   Following the GraLoRA analysis, we decompose one training step into three stages. With a $b \times b$ partition and sequence length $T$, each column slice $X_j \in \mathbb{R}^{(n/b) \times T}$ is first projected to $Z_{ij} = A_{ij} X_j \in \mathbb{R}^{(r/b) \times T}$. This costs $(2n/b - 1)(r/b)T$ FLOPs per block, so across $b^2$ blocks we obtain

$$\text{FLOPs}_1 = (2n - b)rT.$$

The reconstruction stage multiplies $Z_{ij}$ by $B_{ij} \in \mathbb{R}^{(m/b) \times (r/b)}$, yielding $Y_{ij} = B_{ij} Z_{ij} \in \mathbb{R}^{(m/b) \times T}$ at the same per-block cost, hence

$$\text{FLOPs}_2 = (2m - b)rT.$$

Unlike GraLoRA, BHRA applies a multiplicative mask $H_{ij} = W_{0,ij} \odot C_{ij}$. If this mask is evaluated online, it introduces $(mn/b^2)T$ element-wise products. Refreshing $C_{ij} = B_{ij} A_{ij}$ once per step costs $2(m/b)(n/b)(r/b)$ FLOPs per block, or $2mnr/b$ overall. Summing all terms yields

$$\text{FLOPs}_{\text{BHRA}}^{\text{(train)}} = 2r(m+n)T - 2brT + \frac{mn}{b^2}T + \frac{2mnr}{b}. \tag{14}$$

The $2r(m+n)T$ term matches HiRA's rank-dependent scaling; the remaining corrections stem from the block partition and vanish when $b = 1$.

**Relation to GraLoRA and inference cost.**   Setting $W_{0,ij} = \mathbf{1}$ recovers GraLoRA with $k = b$, where the Hadamard stage disappears and the expression above reduces to the classic $(2n - k)rT + (2m - k)mT + (k - 1)mT$ form. In BHRA we precompute the masks $H_{ij}$ and fold them into $W_{0,ij}$ before deployment, so inference evaluates only the two low-rank GEMMs per block:

$$\text{FLOPs}_{\text{BHRA}}^{\text{(adapter)}} = (2n - 1)rT + (2m - 1)rT \approx 2r(m+n)T, \tag{15}$$

identical in order to HiRA and LoRA.

**Memory footprint.**   We separate forward activations from persistent parameters. Like LoRA/HiRA, BHRA stores the projected features $Z_{ij}$ to backpropagate through $A_{ij}$; these consume $rT$ elements independent of $b$. The reconstructed tensors $Y_{ij}$ can be released after the Hadamard modulation. The block masks $C_{ij}$ add $mn$ cached values shared across all tokens and reused for gradients, mirroring GraLoRA's expanded latent space. Without checkpointing the peak layer memory is $rT + mn/b^2$ elements—$rT$ for the stored projections and $mn/b^2$ to keep the block masks active during the forward/backward pair. With gradient checkpointing, $Y_{ij}$ and $H_{ij}$ are recomputed on demand, so the peak requirement shrinks to the $rT$ latent cache plus the persistent masks. Because $r \ll m, n$, the additional $mn/b^2$ term remains negligible relative to the base model activations, keeping BHRA in the same empirical regime as HiRA and GraLoRA.

Table 5: Hyperparameter settings for training Llama-3.2 1B and 3B on COMMONSENSE170K, and Mistral-7B and Gemma-2 9B on MetaMathQA.

|  | Llama-3.2 1B / 3B | Mistral-7B / Gemma-2 9B |
| --- | --- | --- |
| Optimizer | AdamW | AdamW |
| Batch size | 6 | 1 |
| Max. Seq. Len | 256 | 512 |
| Grad Acc. Steps | 24 | 32 |
| Epochs | 2 | 1 |
| Dropout | 0.05 | 0 |
| Learning Rate | $2 \times 10^{-3}$ | $2 \times 10^{-3}$ |
| Target Modules | q_proj, k_proj, v_proj, o_proj, up_proj, down_proj | |
| LR Scheduler | Linear | Cosine |
| Warmup Ratio | 0.02 | 0.02 |

## C   EXPERIMENTAL DETAILS

We implement all models in PyTorch (Paszke et al., 2019) with HuggingFace Transformers (Wolf et al., 2020). Experiments run on 8 NVIDIA A100 80 GB GPUs, and we initialize base models

in torch.bfloat16 to reduce memory consumption. Every configuration is trained with the AdamW optimizer (Loshchilov & Hutter, 2017), and we report the mean performance across five random seeds (42, 2025, 2024, 2023, 2022).

We configure Llama-3.2 1B, Llama-3.2 3B, Mistral-7B, and Gemma-2 9B using the hyperparameters in Table 5. We conduct a sweep over learning rates and scaling factors to identify optimal settings for each model-task pair. BHRA generally performs better with slightly higher learning rates compared to LoRA, and we recommend initiating hyperparameter sweeps in that range.

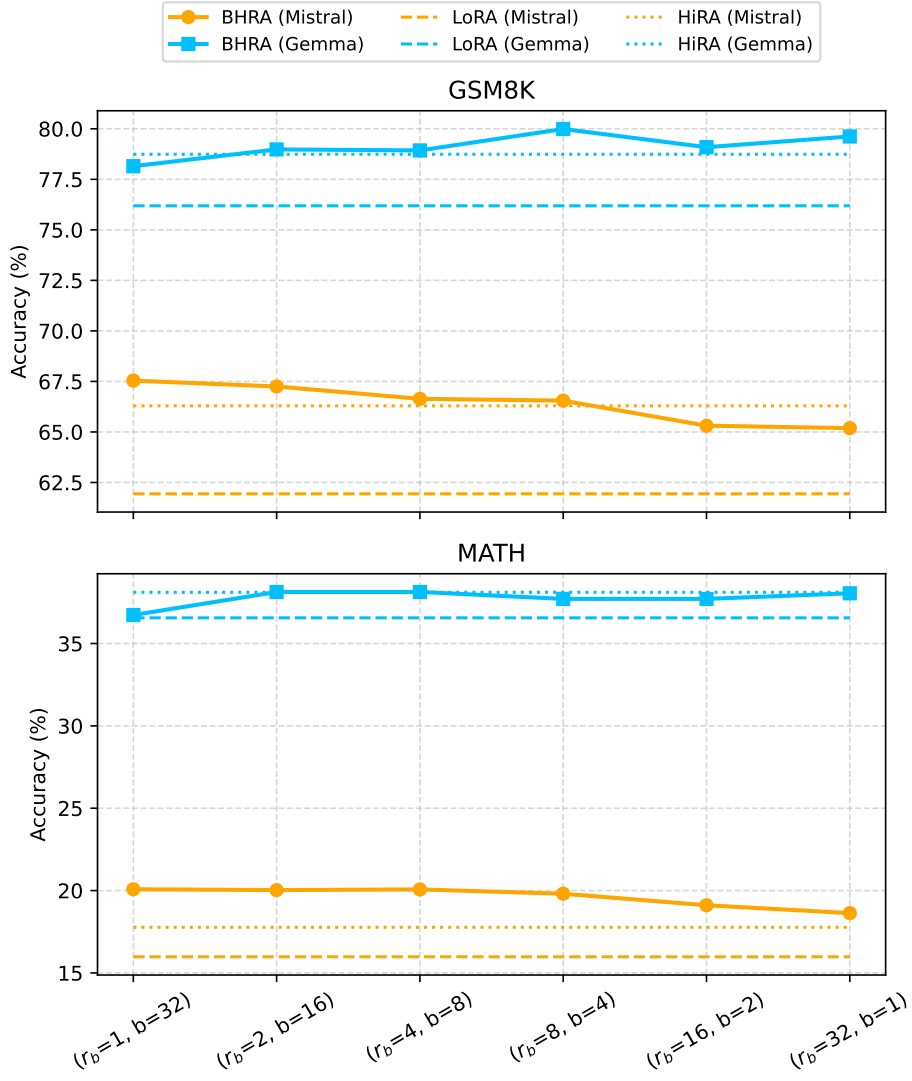

Figure 9: Performance of BHRA, LoRA, and HiRA under a fixed budget ($r \times b = 32$) on arithmetic tasks.

While we adopt most settings from prior work (Hu et al., 2023), we run targeted learning-rate sweeps to tune performance. For baselines we replicate the experimental protocols from LoRA (Hu et al., 2022), DoRA (Liu et al., 2024), HiRA (Huang et al., 2025), and ABBA (Singhal et al., 2025) To contextualize BHRA's behavior, we summarize their key ideas below:

- LoRA: Freezes pretrained weights and injects a pair of rank-$r$ matrices whose product forms a low-rank update, yielding parameter-efficient adapters.
- DoRA: Decouples the update direction and magnitude so the adapter can match high-rank structure while retaining LoRA's parameter count.
- HiRA: Modulates the LoRA update with a Hadamard product with the frozen weights, amplifying the attainable rank in proportion to $\mathrm{rank}(W_0)$.
- ABBA: Learns two independent low-rank factors whose Hadamard combination produces a dense, high-rank update without referencing $W_0$.

## D    EXTENDED EXPERIMENTS

Figure 9 extends the block sweep to GSM8K and MATH. For both base models the BHRA curve peaks at $r_b = 4$ ($b = 8$), matching the commonsense study: moving left to $r_b = 1$ introduces many tiny blocks that slightly underfit (~2 accuracy points on GSM8K for Mistral), while moving right to $r_b \geq 16$ collapses block diversity and erodes the gains (~1–2 points on both datasets). Across the entire sweep BHRA maintains a margin over LoRA and HiRA—Gemma retains a ~3 point advantage on MATH and ~4 points on GSM8K, and Mistral stays 1–2 points ahead except for the largest blocks—highlighting that the balanced $r_b = 4, b = 8$ setting offers the best trade-off between per-block expressivity and diversity on arithmetic reasoning as well.

## E    ETHICS STATEMENT

This work complies with the ICLR Code of Ethics. Our study focuses on low-rank adaptation techniques for large language models. All datasets used in our experiments (i.e., eight commonsense resoning sub-tasks, and arithmetic resoning dataset including MetaMathQA, MATH and GSM8K) are publicly available and have been widely used in prior research. No private or personally identifiable information is included. Since no human subjects were involved, IRB approval was not required. We acknowledge that adapting large-scale models may raise potential ethical concerns, such as misuse for generating harmful or biased content. Our intention is purely to advance the efficiency and accessibility of model fine-tuning for research purposes. We encourage responsible and fair use of our methods and note that mitigation strategies against misuse (e.g., filtering, safety alignment) should be applied when deploying adapted models in real-world scenarios.

## F    THE USE OF LARGE LANGUAGE MODELS (LLMs)

We used large language models (LLMs) to assist with the linguistic polishing of this paper. The models were not involved in designing the methodology, conducting experiments, or drawing conclusions.

