# OpenReview forum: "Blockwise Hadamard high-Rank Adaptation for Parameter-Efficient LLM Fine-Tuning"
_ICLR.cc/2026/Conference — ICLR 2026 Conference Withdrawn Submission_

### Official Review · Reviewer_soYW · 2025-10-30

**Soundness:** 3
**Presentation:** 2
**Contribution:** 2
**Rating:** 2
**Confidence:** 3

**Summary:**

The paper proposes a block-wise Hadamard-style PEFT algorithm, that improves the rank upperbound of the update matrix compared to the previous HiRA method. The authors evaluate the algorithm on commonsense and math reasoning benchmarks with

After carefully reading the paper, I find the novelty of the proposed BHRA method to be limited, and I’m not fully convinced that:

1. The effectiveness of the BHRA method is due to its large rank capacity.
2. BHRA is generally a more effective algorithm than other baselines, even on more challenging benchmarks and large-scale settings.

For this paper to make a stronger impact, it should either present a methodological innovation or provide robust and convincing evidence of its effectiveness for practical (especially scalable) usage. Therefore at this stage I recommend rejecting the paper. However, I am willing to re-evaluate after getting more insights from the discussion with the authors.

**Strengths:**

1. I appreciate the fact that author align the compute budget of the proposed method with baseline methods. The comparison is fair and rigorous, which makes the results more convincing.
2. I also appreciate the authors for sharing the source code, which have better transparency and reproducibility.
3. The authors provide detailed hyperparameter searching and show some insights on the choice of the $\beta$ parameter for BHRA method, which helps the reader to understand more about the core principles of the method.

**Weaknesses:**

1. The motivation of the paper seems mild and not persuasive. The proposed BHRA method presents a way to improve the rank upper bound of the weight update of the HiRA method. However, the motivation is not justified as experimental results don't seem to suggest that a larger rank upper bound is always better.
2. The layout of some figures is not reading-friendly and can be improved. For example, the lines in Figs. 3 and 8 are horizontally indistinguishable, which undermines the main message of the paper. The authors could consider reorganizing the plots so that the horizontal trend is clearer.
3. The tasks chosen to evaluate the BHRA algorithm are not comprehensive. The chosen tasks (GSM8K, MATH, etc.) do not seem challenging enough that require a larger rank upper bound. The authors may want to choose (more complex) datasets that potentially need a larger capacity (rank) to truly demonstrate the effectiveness of the BHRA method.

**Questions:**

1. The main motivation of the paper is that the weight update of BHRA has a higher rank upper bound than HiRA ($br_0r_b$ compared to $r_0r$). However from the experiments it seems that larger rank upper bound does not always have better performance (Figs. 9 & 10). Does the authors have an idea why this
2. The authors show that the proposed BHRA method incurs larger effective rank than the HiRA method. I am confused how a larger rank upper bound connects to larger effective rank. Does the effective rank of BHRA exceeds the upper bound of HiRA? If not, given that the weight updates of both HiRA and BHRA is not large enough, what structural advantages of BHRA causes its weight update to have larger effective rank?
3. In the paper the authors claim that when setting $b = 1$, BHRA recovers HiRA. However from a few of the results (e.g. Fig. 8, Fig. 9), when $r_b = 32, b = 1$, we don't see the performance of BHRA and HiRA matching. Could the authors explain what might cause this discrepancy?
4. The evaluation setup presented in the paper seems not challenging enough to reach the large rank capacity of BHRA, as recent works have shown that even rank-1 updates [1] can reach good enough performance on these datasets. Have the authors explored more challenging benchmarks to evaluate the BHRA method? Such as comparing rank-1 LoRA and BHRA on LLaMA-3.1-8B on GSM or MATH using e.g. GRPO, as in [1]? Or training Qwen-2.5-3B (or larger) model on GSM with DAPO [2]
5. Hadamard-based adapter methods have been existing for a while, yet we don’t have a clear vision on why this method works well (at least better than other LoRA-based method). Therefore it would be more meaningful if there are discussions on what advantages this method has (more than the mere fact that gives a larger rank upper bound). Does the authors have unique opinions on why Hadamard-based methods work, other than the larger rank upper bound of the weight update?

[1] LoRA Without Regret. https://thinkingmachines.ai/blog/lora/

[2] QeRL: Beyond Efficiency -- Quantization-enhanced Reinforcement Learning for LLMs https://arxiv.org/pdf/2510.11696

---

### Official Review · Reviewer_RFEt · 2025-11-01

**Soundness:** 2
**Presentation:** 3
**Contribution:** 1
**Rating:** 2
**Confidence:** 4

**Summary:**

This paper proposes Blockwise Hadamard High-Rank Adaptation (BHRA), which is a parameter-efficient fine-tuning method that splits each weight matrix of an LLM into blocks and applies Hadamard-style (element-wise) modulation separately in each block, instead of globally like HiRA. This preserves LoRA-level parameter and FLOP cost but lets each block adapt with its own high-rank update, which maintains much higher effective rank and local flexibility. As a result, under the same budget, BHRA matches or may beat strong PEFT baselines (LoRA, DoRA, HiRA, ABBA) on commonsense and arithmetic reasoning across Llama-3.2, Mistral-7B, and Gemma-2 9B.

**Strengths:**

1. This paper is well written.
2. The BHRA method is intuitive and easy to understand.

**Weaknesses:**

**If the authors address the issues in the Weaknesses and Questions sections, I will consider to increase the score.**

1. In my opinion, the main issue is that I feel BHRA lacks novelty. Using a Hadamard-product adapter to increase the adapter’s rank was already proposed in HiRA [1] and ABBA[2], and the idea of splitting the weight matrix into sub-blocks and applying PEFT has also appeared in GraLoRA[3]. So overall I think BHRA is relatively incremental. In addition, based on the results in Tables 2 and 3, BHRA and ABBA perform similarly. For example, in Table 2 BHRA and ABBA are very close on Llama-3.2 1B, and in Table 3 BHRA is actually worse than ABBA on Gemma-2 9B.

2. I’m a bit confused about whether increasing the adapter rank has a necessary or sufficient relationship with better fine-tuning performance. Even though the Hadamard product may increase the adapter’s rank, in Figures 2 and 6 the rank of LoRA is extremely low and much smaller than BHRA’s, yet the fine-tuning performance gap between them is not so large. So I’m not sure whether  increasing adapter rank is really a sound core motivation, but I think it can be one part of analysis for any PEFT method.

3. I also find that the paper is missing comparisons against some important PEFT methods, including weight adapters and memory-efficient sparse fine-tuning methods such as [2][3][4][5][6]. It would be better if the authors could include comparisons to these approaches.

4. Minor:  It seems that the example formula for $\Delta W$ in Figure 1(c) is incorrect.

[1] Huang, Qiushi, et al. "HiRA: Parameter-efficient hadamard high-rank adaptation for large language models." The Thirteenth International Conference on Learning Representations. 2025.

[2] Singhal, Raghav, et al. "ABBA: Highly Expressive Hadamard Product Adaptation for Large Language Models." arXiv preprint arXiv:2505.14238 (2025).

[3] Jung, Yeonjoon, et al. "GraLoRA: Granular Low-Rank Adaptation for Parameter-Efficient Fine-Tuning." arXiv preprint arXiv:2505.20355 (2025).

[4] Meng, Fanxu, Zhaohui Wang, and Muhan Zhang. "Pissa: Principal singular values and singular vectors adaptation of large language models." Advances in Neural Information Processing Systems 37 (2024): 121038-121072.

[5] Zhang, Fangzhao, and Mert Pilanci. "Spectral adapter: Fine-tuning in spectral space." arXiv preprint arXiv:2405.13952 (2024).

[6] Ansell, Alan, et al. "Scaling sparse fine-tuning to large language models." arXiv preprint arXiv:2401.16405 (2024).

[7] Liu, Zihang, et al. "LIFT the Veil for the Truth: Principal Weights Emerge after Rank Reduction for Reasoning-Focused Supervised Fine-Tuning." arXiv preprint arXiv:2506.00772 (2025).

**Questions:**

1. Could you also add the corresponding ABBA case to Figures 2, 4, 5, and 6?
2. I'm confused about Figure 8. If I understand correctly, when $b = 1$, BHRA and HiRA should be the same. Why do the points for BHRA and HiRA in the plot not overlap when $b=1$?

---

### Official Review · Reviewer_AMWg · 2025-11-02

**Soundness:** 3
**Presentation:** 3
**Contribution:** 2
**Rating:** 6
**Confidence:** 3

**Summary:**

The paper introduces Blockwise Hadamard High-Rank Adaptation (BHRA), a parameter-efficient fine-tuning (PEFT) method that extends Hadamard-style adapters (HiRA, ABBA) by applying localized blockwise multiplicative modulation on pretrained weight matrices. The idea is to partition each matrix into b×b blocks and apply HiRA-style low-rank updates independently per block. This aims to overcome the “global modulation” limitation of HiRA while maintaining LoRA-level parameter and FLOP budgets. The paper provides a detailed theoretical rank analysis, efficiency bounds, and empirical evaluations across commonsense and arithmetic reasoning benchmarks (Llama-3.2, Mistral-7B, Gemma-2 9B), showing consistent improvements over LoRA and HiRA.

**Strengths:**

* **Clear motivation:** The paper convincingly identifies the global coupling issue in HiRA and demonstrates the benefits of localized modulation through stable-rank and spectral analyses.
* **Strong theoretical grounding:** The rank-bound derivation (( \text{rank}(\Delta W_{BHRA}) \leq b,r_0,r )) elegantly generalizes HiRA’s ( r_0 r ) bound, providing a clear conceptual contribution.
* **Comprehensive empirical evaluation:** The experiments are well-controlled and reproducible, with comparisons against relevant baselines (LoRA, DoRA, HiRA, ABBA).
* **Good diagnostics:** The analysis of singular value spectra, effective rank, and block-Gini metrics is insightful and supports the core claims.

**Weaknesses:**

---

### **Weaknesses / Questions**

* **Relation to Kronecker-LoRA (Kron-LoRA):**
  The proposed BHRA shares conceptual similarities with Kronecker-factorized low-rank adapters, where a Kronecker product or structured tensorization is used to decompose ( \Delta W ) into localized, low-dimensional interactions.

  * Please clarify how BHRA differs *mathematically and functionally* from Kronecker-LoRA or other Kronecker-structured variants (e.g., in factorization form, rank behavior, or computational scaling).
  * Both Kronecker and blockwise Hadamard schemes introduce structured factorization that may implicitly increase the effective rank. It would help to discuss whether BHRA’s blockwise Hadamard modulation can be viewed as a *Hadamard–Kronecker hybrid* and how its expressivity compares under equal parameter count.

* **Comparison fairness:**
  While the experiments control for rank and parameter budget, it would be valuable to include a Kronecker-LoRA baseline (or cite results) to establish that BHRA’s gains are not simply due to block-structured factorization.

* **Complexity analysis:**
  The FLOP derivation is detailed, but a numerical estimate (e.g., wall-clock overhead vs HiRA or LoRA on 7B models) would help readers assess practical efficiency.

* **Scope of evaluation:**
  The benchmarks focus on reasoning tasks; including at least one *generation-heavy* task (e.g., summarization or dialogue) could test whether the block-local structure generalizes beyond reasoning.

---

**Questions:**

plz see the above weaknesses.

---

### Official Review · Reviewer_G8YM · 2025-11-02

**Soundness:** 3
**Presentation:** 3
**Contribution:** 3
**Rating:** 6
**Confidence:** 4

**Summary:**

The paper proposes Blockwise Hadamard high-Rank Adaptation (BHRA), a parameter-efficient fine-tuning method that splits each weight matrix into a grid of blocks and applies element-wise modulation inside each block using a learned low-rank factor. This local modulation keeps the pretrained weight structure while avoiding the global coupling and spectral collapse observed in HiRA, so the adapted matrix attains higher effective rank under the same parameter and compute budget as LoRA and HiRA. The authors provide a rank upper bound that scales with the number of blocks, and show empirically on Llama, Mistral, and Gemma models that BHRA improves accuracy on commonsense and math benchmarks and produces spectral profiles closer to full fine-tuning. A simple configuration with total rank 32 (for example, eight blocks per side with rank four per block) and focusing updates in FFN layers works reliably across tasks.

**Strengths:**

(1) Under the same parameter budget, BHRA outperforms LoRA and HiRA on commonsense and math tasks, and its overall performance is comparable to full-parameter fine-tuning.
(2) In the Analysis section, the authors provide a detailed examination of singular values, validating the reasons for BHRA’s superior performance.

**Weaknesses:**

(1) All experiments were conducted with r=32. Considering that r=8/16 are still commonly used hyperparameters, should the authors supplement additional experimental results to demonstrate the robustness of their method?

(2) While the authors theoretically analyze the training cost in Appendix B, they do not provide empirical comparisons of actual training time and memory usage against methods like LoRA in the experimental results.

(3) Could BHRA potentially increase training instability ?

(4) Only regular square grids are explored. It is unclear whether alternative partitions (rectangular blocks, per-layer b, data-driven grouping) could do better, and how robust results are to the choice of block boundaries.

**Questions:**

Please see the weakness part

---

### Note · Authors · 2025-11-30

I have read and agree with the venue's withdrawal policy on behalf of myself and my co-authors.